# A Cost-Aware DNN-Based FDI Technology for Solenoid Pumps

Suju Kim, Ugochukwu Ejike Akpudo 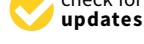 and Jang-Wook Hur *

Department of Mechanical Engineering, Department of Aeronautics, Mechanical and Electronic Convergence Engineering, Kumoh National Institute of Technology, 61 Daehak-ro, Yangho-dong, Gumi 39177, Korea; suju0708@gmail.com (S.K.); akpudougo@gmail.com (U.A.E.)
* Correspondence: hhjw88@kumoh.ac.kr

**Abstract:** Fluid Pumps serve a critical function in hydraulic and thermodynamic systems, and this often exposes them to prolonged use, leading to fatigue, stress, contamination, filter clogging, etc. On one hand, vibration monitoring for hydraulic components has shown reliable efficiencies in fault detection and isolation (FDI) practices. On the other hand, signal processing techniques provide reliable FDI parameters for artificial intelligence (AI)-based data-driven diagnostics (and prognostics) and have recently attracted global interest across different disciplines and applications. Particularly for cost-aware systems, the choice of diagnostic parameters determines the reliability of an FDI/diagnostic model. By extracting (and selecting) discriminative spectral and transient features from solenoid pump vibration signals, accurate diagnostics across operating conditions can be achieved using AI-based FDI algorithms. This study employs a deep neural network (DNN) for fault diagnosis after a correlation-based selection of discriminative spectral and transient features. To solve the problem of hyperparameter selection for the proposed model, a grid search technique was employed for optimal search for parameters (number of layers, neurons, activation function, weight optimizer, etc.) on different network architectures.The results reveal the high accuracy of a three-layer DNN with *ReLU* activation function, with a test accuracy of 99.23% and a minimal false alarm rate on a case study.

**Keywords:** condition monitoring; fault diagnosis; feature extraction; feature selection; deep neural network

## 1. Introduction

Solenoid pumps serve a critical function in hydraulic and/or thermodynamic systems, supplying fluid to the desired location at the desired pressure. Due to prolonged operation, these pumps are exposed to various failure modes such as fluid contamination, electrical/mechanical stress, fatigue, and filter clogging [1]. As these failures greatly affect productivity and uninterrupted manufacturing/production processes, the need for failure prognostics and health management (PHM) technologies have become increasingly high. Against the limitations of traditional PHM methodologies, which rely greatly on physics-of-failure (PoF) and expensive assumptions during dynamic modelling, AI-based PHM technologies offer better dynamic modelling and accuracy for predictive maintenance [2,3]. From a broader perspective, the effectiveness of data-driven condition-based maintenance (CBM), which constitutes PHM modules at its core for predictive maintenance, relies on the use of sensor measurements; however, these measurements (operation data) produce complex big data, from which FDI can be implemented using befitting AI-based solutions. By so doing, equipment health status can be properly monitored [4,5].

Related research studies on solenoid pump FDI and prognostics are actively in progress, with several studies employing diverse data-driven diagnostics (and prognostics) technologies. For instance, Akpudo and Hur [2] extracted Mel frequency cepstral coefficients (MFCC) as fault features and, with the radial basis function support vector machine (SVM), isolated several operating conditions. This, amongst many other AI-based methods,

has been reported with remarkable results; however, considering the complexity in the operational behavior of solenoid pumps and the increasing concern for computational costs associated with real-time applications, this study aims to identify key fault parameters from vibrational measurements (via signal processing methods) as inputs to an artificial neural network (ANN)-based classifier for FDI.

The effectiveness/efficiency in fault parameter identification by a diagnostic model is one of the major deciding criteria for assessing the reliability of FDI technologies. Often, vibration measurements are corrupt with background noise and are non-stationary. This ushers in the need for transient and spectral characteristic information extraction for accurate diagnosis from an empirical standpoint [6]. To achieve these solutions, signal processing techniques offer a reliable paradigm. Among these techniques are the time-domain, frequency-domain, and the more expensive time-frequency-domain signal processing techniques. Although time-frequency-domain techniques such as the MFCCs, continuous wavelet transform (CWT), empirical mode decomposition (EMD), variable mode decomposition (VMD), and short-time Fourier transform (STFT), etc. are robust for transient and spectral feature extraction (and for most de-noising problems), the costs associated with their computation process has been a major concern for their real-time/instant useability. For instance, the iterative process of EMD [7], strenuous choice of window function for STFT [8], and the exhaustive search for optimal wavelet functions by the CWT [9] are obvious concerns in cost-aware systems. Consequently, the use of statistical features from time-domain and frequency-domain (following a Fourier transform) provide a reliable, cost-efficient avenue for discriminative fault parameter (feature) extraction.

Although statistical features are quite reliable as fault features for FDI, the need for feature assessment based on discriminative importance cannot be overemphasized [9]. By assessing fault features, key diagnostic features can be identified to further minimize the computational costs associated with the FDI technology while also enhancing the diagnostic prowess of the fault isolator (AI-based classifier). Many filter-based, wrapper-based, and hybrid feature selection methods abound with their respective pros and cons [10]; however, from a realistic standpoint, filter-based methods, which are purely unsupervised in architecture, are more efficient than their wrapper-based counterparts, which indirectly aim to satisfy the classifier's fitness function (and not necessarily to identify discriminative features in an unbiased manner). Interestingly, hybrid methods combine these methods for much better results, but the computational costs and strenuous parameterization issues associated with their computation process usually limit their effectiveness for cost-aware applications [10]. Subsequently, the unsupervised architecture, ease-of-use, and empirically accessibility associated with the filter-based methods make them more preferable for real-time/cost-aware applications.

Recent advances in data-driven FDI suggest the use of ANNs for improved modelling and diagnostic results [11]. These bio-inspired algorithms including convolutional neural network (CNN), recurrent neural network (RNN), deep belief network (DBN), etc. have aroused a global interest in academia and industries. Particularly for fault diagnosis, DNNs and CNNs are quite efficient, with CNNs incorporating automated feature learning capabilities in their architecture while DNNs are mostly employed as classifiers given discriminative labeled inputs [12]. On the downside, the magical defiance to empirical interpretations associated with CNNs and their overfitting tendencies have encouraged the use of DNNs, which come with comparatively less computational costs and parametrization [13]. On other domains such as handling identity switches, the authors of [14] employed a multi-player tracker incorporating deep player identification for producing identity-coherent trajectories. The results therein also reveal the superior efficiencies of deep models. Interestingly, ongoing research studies in the domain have revealed the efficiency of standalone deep models not only for various tasks but also for optimizing and improving them for even better results across diverse applications [14–18]. Consequently, this study employs a DNN-based FDI approach for solenoid pumps.

The major contribution of this study is the proposal of a cost-efficient FDI technology for solenoid pumps based on spectral and transient diagnostic information. The proposed model was validated on an actual test-bed currently situated in the Defense Reliability Laboratory, Department of Mechanical Engineering, Kumoh National Institute of Technology, Republic of Korea. The rest of the paper is organized as follows: Section 2 presents the theoretical overview of the key phenomena employed in this study, while Section 3 presents the proposed FDI model. Section 4 presents the experimental analysis and results, while Section 5 concludes the paper.

## 2. Theoretical Framework

### 2.1. Salient Feature Extraction

Raw vibrational signals from real applications are usually non-stationary in nature and may be contaminated with background noise. Extracting salient features from these signals remains an ongoing challenge for accurately understanding the underlying dynamics of targeted systems; even so, some statistical feature extraction techniques in the time domain and frequency domain provide a reliable avenue for accurate condition monitoring. Although these features such as Kurtosis, skewness, clearance factor, spectral centroid, etc. all provide unique insights from input variables (vibration signals), using all features for diagnostics may have dire effects on computational resources and the classifier so there is a need for salient feature selection.

Against the earlier mentioned limitations of wrapper-based methods which indirectly aim to satisfy the classifier's fitness function (and not necessarily for discriminative feature selection in an unbiased manner), several filter-based methods, which are purely unsupervised, are more efficient and are more resource-friendly. A popular, convenient (and easy-to-use), and fast feature selection algorithm is the Pearson's correlation algorithm [19], which computes the linear dependence between two continuous variables and returns a value between the range −1 (*negative correlation*) and +1 (*positive correlation*). This is obtained using Equation (1).

$$\rho_{X,Y} = \frac{\text{cov}(X,Y)}{\sigma_X \sigma_Y} \tag{1}$$

where $\sigma_X$ and $\sigma_Y$ are the standard deviation of $X$ and $Y$, respectively, and $\text{cov}(X,Y)$ is the covariance.

### 2.2. Deep Neural Network (DNN)

Recently, the dominance of AI has ushered in the use of machine learning (ML) algorithms for diverse solutions and with the recent advancements in technology, the more robust DL methods have become remarkably preferable for diverse tasks. Among the ML algorithms is the multi-layer perceptron (MLP)—a feed-forward neural network (FFNN), which typically consists of three basic structures: an input layer, a hidden layer, and an output layer [20].

The learning procedure of a typical MLP (or DNN) consist of three major steps—forward propagation, cost function minimization, and backward propagation. To explain in detail, the learning process starts from the nodes in the input layer, where each node exports its input value to the next layer via a weighted forward propagation process with an activation function. During the backward propagation (of weights) process, the learning is achieved by a cost minimization (reducing the squared errors between the predicted and actual labels by adjusting the learned weights) by stochastic gradient descent (SGD) [20,21].

For better learning, a typical MLP can be improved to form a DNN by increasing the number of hidden layers to two or more hidden layers as illustrated in Figure 1 with the layers comprising several nodes—$m$ nodes in the input layer, $p$, $q$, and $r$ nodes in the first, second, and third hidden layers, respectively, and $n$ nodes in the output layer.

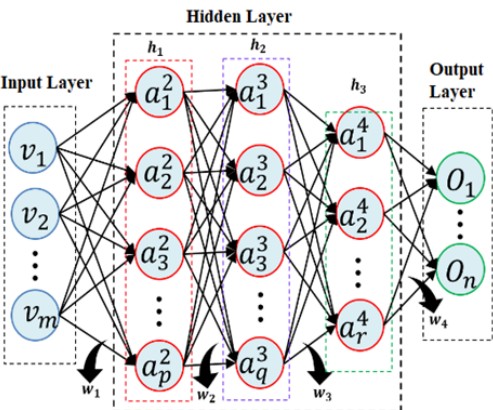

**Figure 1.** A DNN with three hidden layers (MLPs).

To better explain using Figure 1, considering the nodes $\{a_1^4, a_2^4, \ldots, a_r^4\}$ in the hidden layer $h_3$ and the nodes in the output layer $\{O_1, O_2, \ldots, O_n\}$, each of the nodes $a_i^4$ $\{i = 1, 2, \ldots, r\}$ first receive activated outputs from the preceding layer $h_2$ and via a forward propagation process, they simultaneously compute activated outputs to the outer layer's nodes $O_i$ $\{i = 1, 2, \ldots, n\}$. The learning (supervised) process is achieved via the back-propagation of the weights $w_4$ to minimize the cost function by SBD, *Adam* [22], and/or any of the quasi-Newton methods [23].

Empirically, the inputs $\mathbf{0}_{i\_in}$ provided by the node $a_i^4$ are received by the nodes $O_i$ given by the sum of the activated outputs of $a_i^4$ multiplied by the corresponding connection weight matrix $w_4$ using Equation (2).

$$\mathbf{0}_{i\_in} = \sum_{i=1}^{r} w_4 * A[i] \tag{2}$$

where $A[i]$ is the activated outputs of the nodes in $h_3$.

The output $\mathbf{0}_{i\_out}$ from each of the output nodes $O_i$ is obtained by passing the inner product $\mathbf{0}_{i\_in}$ through a nonlinear activation function $f$ using Equation (3):

$$\mathbf{0}_{i\_out} = f(\mathbf{0}_{i\_in}) \tag{3}$$

where the choice of $f$ ranges across popular functions such as *Sigmoid*, *Tanh*, rectified linear unit (*ReLU*), and *Leaky ReLU*, among which the ReLU is the most effective for classification problems as it returns the corresponding output value for positive values, whereas for negative input values, it returns a zero value [21].

The automatic (supervised) learning process of DNN by gradient descent enables minimizing the squared error in the predicted outputs and the actual target labels via a back-propagation of weights using Equation (4):

$$E = (y - \mathbf{0}_{i\_out})^2 \tag{4}$$

where $E$ is the prediction error (cost function) and $y$ is the desired output label.

More often, the *ReLU* activation function is preferred because of its faster learning advantages on DNNs and because it returns the corresponding output value for positive values whereas, for negative input values, it returns a zero value [21]; ideally, most input variable-extracted features are greater than zero, and *ReLU* is often preferred. On the other hand, the choice of weight optimization method differs for different kinds of problems. Ideally, the SGD works quite well for most problems; however, in *Adam*—an improved SGD-based optimizer with better efficiencies—was proposed in 2015 and has gained wider popularity over the standard SGD optimization technique due to its superior efficiencies (faster convergence and improved validation scores) on relatively large datasets. Apart

from these methods, quasi-Newton methods are also popular and are relatively more efficient for small datasets.

To ensure that the right parameter combination is obtained for a DNN architecture, it is often unwise to randomly choose parameters as this may be realistically unreliable. In addition to a random selection of parameters, more reliable methods exist for solving optimal parameter selection including the exhaustive search method, meta-heuristic optimization method, and Grid search methods. Considering the cost-awareness of the proposed diagnostics framework, the grid search method suffices because both the exhaustive and meta-heuristic optimization methods are very computationally expensive.

### 3. Overview of the Proposed FDI Model

Figure 2 shows an illustration of the proposed FDI framework consisting mainly of the key sections: statistical feature extraction, filter-based feature selection, fault isolation by the DNN, and performance evaluation.

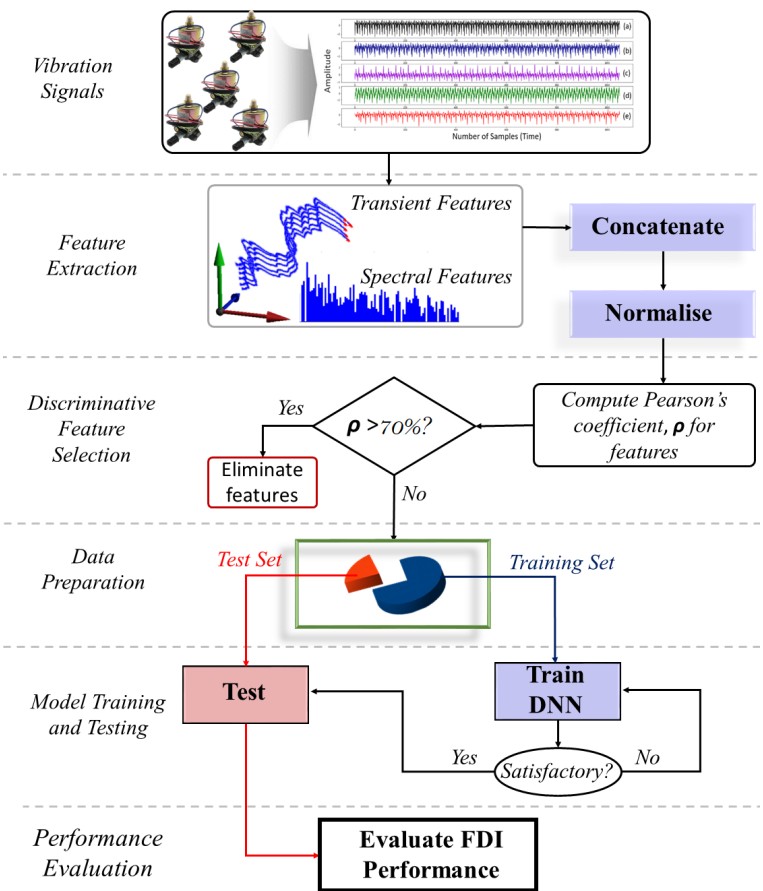

**Figure 2.** Proposed system model.

The comprehensive approach of the proposed FDI model results in the extraction of transient and spectral diagnostic features that are collectively useful in fault detection; however, the selection of discriminative features provides a more robust and cost-efficient diagnostic framework.

### 3.1. Feature Extraction and Selection

Following Figure 2, from the vibration signals, transient, and spectral features are extracted (after data cleaning) from the time and frequency domains followed by a post-processing step that entails concatenation and normalization of these features to form a comprehensive feature matrix. The normalization step ensures the feature values are in the

range {0–1}, thereby ensuring that the contribution of each feature to the feature selection module is unbiased. Next, Pearson's correlation algorithm is employed for a filter-based feature selection process whereby features with at least 70% positive correlation value between them are dropped while the uncorrelated features are selected for training using the DNN classifier.

### 3.2. Model Training and Performance Evaluation

These uncorrelated features are labeled according to pump operating conditions and input to the DNN classifier for training. Following a successful training using the training dataset, the FDI performance of the diagnostic model is tested with the test set (created by a similar approach of feature extraction and selection) and evaluated using standard classification/diagnostic performance evaluation metrics defined in Equations (5)–(9) below.

$$\text{Accuracy} = \frac{TP}{TP + FP + TN + FN} \tag{5}$$

$$\text{False Alarm Rate} = \frac{FP}{FP + TN} \tag{6}$$

$$\text{Sensitivity} = \frac{TP}{TP + FN} \tag{7}$$

$$\text{Precision} = \frac{TP}{TP + FP} \tag{8}$$

$$\text{F1 Score} = \frac{2 * \text{Sensitivity} * \text{Precision}}{\text{Precision} + \text{Sensitivity}} \tag{9}$$

Respectively, *TP, FP, TN,* and *FN* are the number of correctly classified samples, number of incorrectly classified samples, number of incorrectly labeled samples (belonging to a class that was correctly classified), and the number of incorrectly labeled samples belonging to a class that was incorrectly classified.

### 4. Experimental Study

The proposed cost-aware FDI technology was employed on a physical setup that consisted of five VSC63 solenoid pumps (manufactured by Korea Control Limited) of which the specifications are summarized in Table 1.

**Table 1.** Solenoid pump specification.

| Model | Rated Voltage | Power Rating | Max Flow | Max Pressure |
|-------|---------------|--------------|----------|--------------|
| VSC63 | 220 V | 25 VA | 17.7 L/h | 17.3 kgf/cm$^2$ |

Each pump was conditioned to operate under different failure conditions while vibration signals were digitally acquired for analysis via high sensitivity accelerometers.

### 4.1. Data Acquisition

The full data acquisition process from the testbed is shown in Figure 3a, while Figure 3b shows an actual view of the test bed currently situated in the Defense Reliability Laboratory, Department of Mechanical Engineering, Kumoh National Institute of Technology, Republic of Korea.

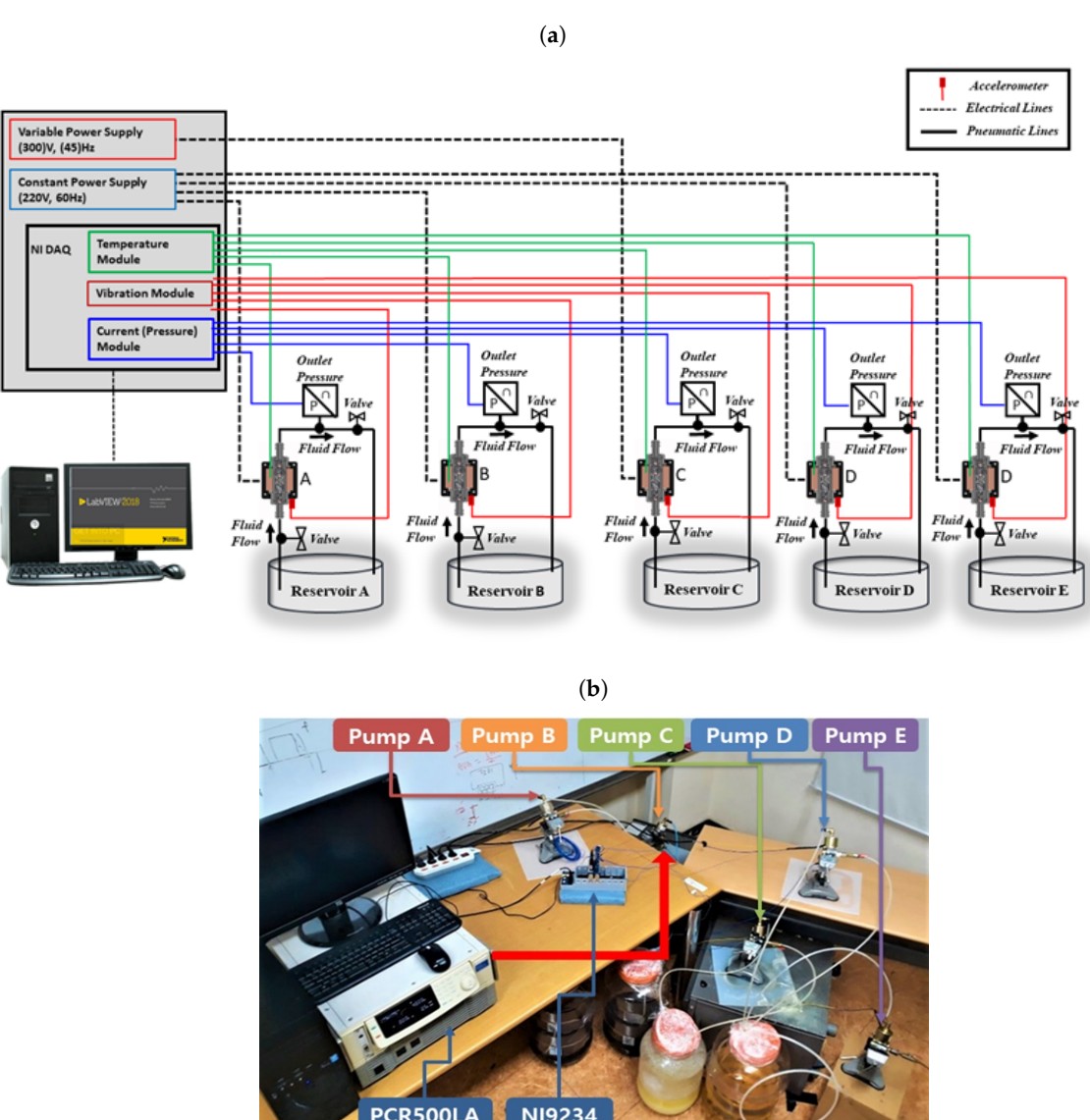

**Figure 3.** Data acquisition process showing (**a**) an illustration of the full data acquisition process and (**b**) actual experimental setup.

The reservoir for Pump A contains diesel mixed with aluminum oxide to simulate filter clogging, while pump B's reservoir contains diesel mixed with SAE40 engine oil to simulate high viscosity. Pump D's reservoir contains a mixture of pump A and pump B's working fluids to simulate an even harsher operating condition, while pump E is operated under the normal operating condition. Apart from pump C, which was powered by a variable AC (300 V, 45 Hz) to simulate an unspecified power supply condition, the pumps were powered by a stable supply of 200 V and 60 Hz via an automatic voltage regulator.

For each of the operating conditions, data collection was performed simultaneously (and consecutively) for four days, and for each day, vibration signals were collected thrice for at least ten (10) min—morning, afternoon, and evening to account for environmental (weather) influence. From the whole data collected, a selected portion (about 40% of the whole dataset) was reserved for testing while the remaining was labelled accordingly for training. It is worth noting that this reserved portion of the data (test data) contains signals for each of the times and for each of the days.

As a summary, Table 2 presents the operating conditions of the pumps labeled A–D.

**Table 2.** Proposed pump operating conditions.

| Label | Input Power | Operating Condition | Failure Mode |
|---|---|---|---|
| A | 220 V, 60 Hz | 3 L Diesel, 300 g $Al_2O_3$ | Filter Clogging |
| B | 220 V, 60 Hz | 2 L Diesel, 2 L SAE40 Engine Oil | High Viscosity |
| C | 300 V, 45 Hz | 2 L Diesel | Unspecified Power |
| D | 220 V, 60 Hz | 2 L Diesel, 2 L SAE40 Engine Oil, 300 g $Al_2O_3$ | Filter Clogging, High Viscosity |
| E | 220 V, 60 Hz | 3 L Diesel | Normal |

High sensitivity accelerometers were installed under each pump to capture vibration signals from the Z-axis (this is the direction of oscillation of the plunger); these sensors were connected to an NI DAQ via the NI 9234 module; and from a LabVIEW environment, the data were digitally collected with a sampling rate of 1000 Hz and stored in "csv" formats.

### 4.2. Experimental Results and Validations

Figure 4 shows a view of the vibration samples for each of the pumps for approximately 10 s and the corresponding power spectra of the signals from which the spectral features are extracted.

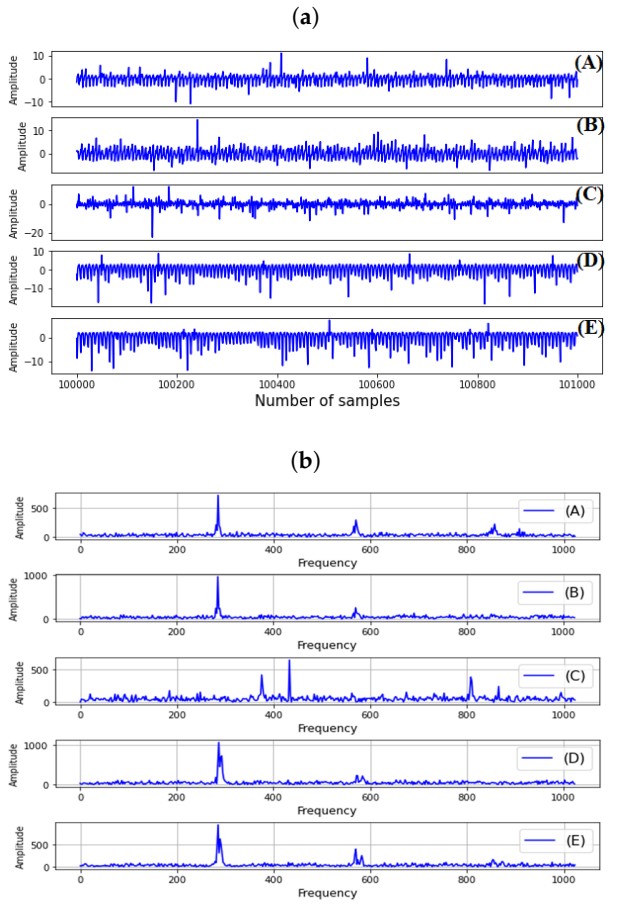

**Figure 4.** A view of the acquired signals from the pumps: (**a**) a view of vibration samples of the pumps and (**b**) power spectra of the pumps' vibration signals.

From Figure 4b, it observed that different pump conditions produce different frequency spikes at different bands. This hints at the discriminance associated with features from frequency spectra. More so, different wave-forms and amplitudes are observed for each of the pumps, as shown in Figure 4a; however, from the raw samples, it may be an uphill battle to identify discriminative characteristics from the signals, hence the need for

feature extraction. Following the earlier described procedure (refer to Figure 2), the vibration signals were cleaned and prepared for feature extraction and selection. The feature extraction and selection processes for the training and test datasets follow the following process: using a window size of 0.1 s (100 samples), the signals for the respective operating conditions were split into uniform portions, and on each of these portions, feature extraction was performed followed by the correlation-based feature discriminative feature selection process explained in Section 2.

As previously mentioned, the model extracts both transient and spectral features, which serve as inputs to the DNN model. To achieve this, concurrently, statistical time domain features are extracted from the raw signals while the spectral features are extracted from the generated power spectra through Fourier transform.

As shown in Figure 4b, there is a clear difference in their respective spectra, and this provides a reliable intuition for possible discrimminace in the signals for FDI; however, being that the signals contain both transient and spectral diagnostic parameters, reliable characteristic information can be captured from both domains. Table 3 summarizes the features (and their respective definitions) employed in this study.

**Table 3.** Extracted features and their definitions.

| Feature Domain | Feature Name | Definition |
|---|---|---|
| Spectral Features | Spectral Centroid | $FC = \frac{\sum_{i=2}^{N} x'_i x_i}{2\pi \sum_{i=1}^{N} x_i^2}$ |
| | RMSF | $RMSF = \sqrt{\frac{\sum_{i=2}^{N}\left(x'_i\right)^2}{4\pi^2 \sum_{i=1}^{N} x_i^2}}$ |
| | Spectral kurtosiss | $SK = \frac{2\sum_{k=0}^{B_L/2-1}\left(|X(k,n)|-\mu_{|X|}\right)^4}{B_L \cdot \sigma_{|X|}^4} - 3$ |
| | Spectral Skewness | $ss = \frac{2\sum_{k=0}^{B_L/2-1}\left(|X(k,n)|-\mu_{|X|}\right)^3}{B_L \cdot \sigma_{|x|}^3}$ |
| Transient Features | Root Mean Square | $X_{rms} = \sqrt{\frac{\sum_{i=1}^{n}(x_i)^2}{n}}$ |
| | Kurtosis | $X_{kurt} = \frac{1}{N}\Sigma\left(\frac{(x_i-\mu)^3}{\sigma}\right)$ |
| | Skewness | $X_{skew} = E\left[\left(\frac{(x_i-\mu)^3}{\sigma}\right)\right]$ |
| | Max | $X_{max} = \max(x_i)$ |
| | Min | $X_{max} = \min(x_i)$ |
| | Crest Factor | $X_{CF} = \frac{x_{max}}{x_{rms}}$ |
| | Shape Factor | $X_{SF} = \frac{x_{RMS}}{\frac{1}{N}\sum_{i=1}^{N}|x_i|}$ |
| | Impulse Fcator | $X_{IF} = \frac{x_{max}}{\frac{1}{N}\sum_{i=1}^{N}|x_i|}$ |
| | Peak-to-peak | $X_{p-p} = x_{max} - x_{min}$ |
| | Median | $\left(\frac{n+1}{2}\right)^{th}$ sample |
| | Mean | $\bar{x} = \frac{1}{n}\left(\sum_{i=1}^{n} x_i\right)$ |
| | Peak factor | $x_{PF} = \frac{x_{max}}{\sqrt{x_s}}$ |
| | Wave Factor | $x_{WF} = \frac{\sqrt{\frac{1}{n}\sum_{i=1}^{n}|x_i|^2}}{\frac{1}{n}\sum_{i=1}^{n}|x_i|}$ |
| | Clearance Factor | $x_{CF} = \frac{x_{max}}{\text{mean }|2x|}$ |
| | Entropy | $x_E = -\sum_{i=1}^{N} P(x_i)\log P(x_i)$ |
| | Zero Crossing Rate | $x_{ZCR} = \frac{1}{2N}\sum_{i=1}^{N}|\sin(x_i) - \sin(x_{i-1})|$ |
| | Mean Crossing Rate | $x_{MCR} = \frac{\sum_{i=2}^{n}|\sin(x_i-\mu)-\sin(x_{i-1}-\mu)|}{2}$ |

Although several other statistical features also provide characteristic information from a signal, the ones employed in this study were chosen based on superior (and popular) significance for reliable empirical validations.

### 4.2.1. Feature Evaluation and Selection

Regardless of the unique significance of each of the extracted features for diagnosis, the influence of highly correlated features on the overall computational process remains a concern for efficient learning and cost awareness. Following a labeling process of the feature set in the range (0–4) corresponding to the pumps (*A–E*), the feature set was fed into the feature selection module for discriminative feature selection. Using Equation (1), the correlation between the features was computed and stored for comparison and selection whereby highly correlated features $\rho_{X,Y} > -0.7$ are dropped while the rest are selected as highly discriminant features. Consequently, 13 out of 21 features were selected and normalized for use by the DNN classifier. Figure 5 shows the correlation plot of the features before and after the feature selection process.

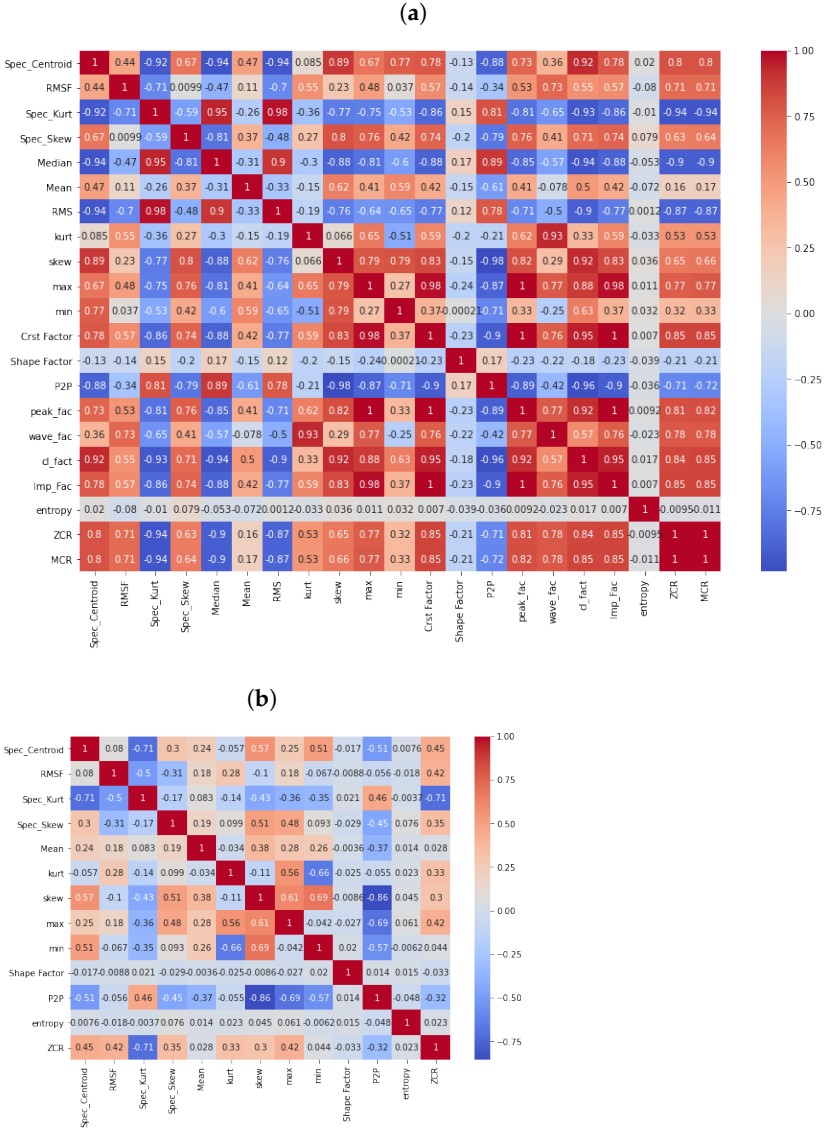

**Figure 5.** Correlation matrix of features: (**a**) all 21 features and (**b**) 13 uncorrelated features.

For instance, as shown in Figure 5a, the features *max, min, crest factor, peak factor, clearance factor*, and *impulse factor* are highly correlated with skewness and have correlation values of 0.79, 0.79, 0.83, 0.82, 0.92, and 0.83, respectively. These are clearly above the 0.7

threshold. A further look at Figure 5a reveals that there are other high correlation values between/amongst some features. Consequently, a feature selection implementation using the proposed approach reveals the correlation matrix of the 13 features in Figure 5b, which not only reduces the feature dimension for cost efficiency but also improves the discriminative power of the feature set for improved classification accuracy. These 13 uncorrelated features are then fed as input to the DNN classifier.

### 4.2.2. DNN-Based Fault Classification

Different DNN architectures with different parameters were designed for diagnostic evaluations. Table 4 summarizes the different DNN architectures and the various parameter options available for use. For each of the DNN architectures summarized in Table 4, there are an additional five neurons in the input and output layers each (each node for the respective pump conditions). These nodes are for accepting the vibration features from the five different pump operating conditions in the input layer while the other five nodes in the output layer are for making the class predictions.

**Table 4.** The different DNN architectures explored by Grid Search.

| Number of Layers (Nodes per Layer) | | Activation Function | Weight Optimizer | Regularization Parameter | Learning Rate |
|---|---|---|---|---|---|
| 3 | (150,100,50) | Tanh, | SGD, | 0.0001, | constant, |
| 3 | (150,100,20) | ReLU, | *adam* | 0.001, | adaptive |
| 3 | (150,50,20) | Logistic, | | 0.01, | |
| 3 | (100,50,20) | Sigmoid | | 0.05 | |
| 2 | (150,100) | | | | |
| 2 | (150,50) | | | | |
| 2 | (150,20) | | | | |
| 2 | (100,50) | | | | |
| 2 | (100,20) | | | | |
| 2 | (50,20) | | | | |
| 1 | (150) | | | | |
| 1 | (100) | | | | |
| 1 | (50) | | | | |
| 1 (20) | | | | | |

The effect of the different activation functions, weight optimizer, regularization parameter, and learning rates on each of the different network architectures ranging from a an MLP (1 layer) to DNN (3 layers) definitely differ. A grid search on a total of 896 unique classifiers was designed by and the grid search algorithm deployed for evaluating their validation accuracies over a five-fold cross validation. It is worth noting that, from the experience of the authors with DNNs, increasing the number of hidden layers beyond three usually tends to over-fit the data and to consume more computational resource. Additionally, the motivation behind the choice of the number of nodes—150, 100, 50, and 20—were purely based on prior experience of choosing the number of neurons in a uniformly descending order. The authors believe that it would be futile to assess all of the possible combinations of all integers (number of nodes) for all possible architectures (number of layers).

Following a grid search on the different DNN (and MLP) architectures summarized in Table 4 with a five–fold cross validation of each of architecture with a 20% validation set (reserved from the training data set) over 500 epochs/iterations with 10 mini batches per epoch, Figure 6 show the parameter sensitivity analysis effect of the different number of layers, nodes, and other other parameters summarized in Table 4 on the validation scores.

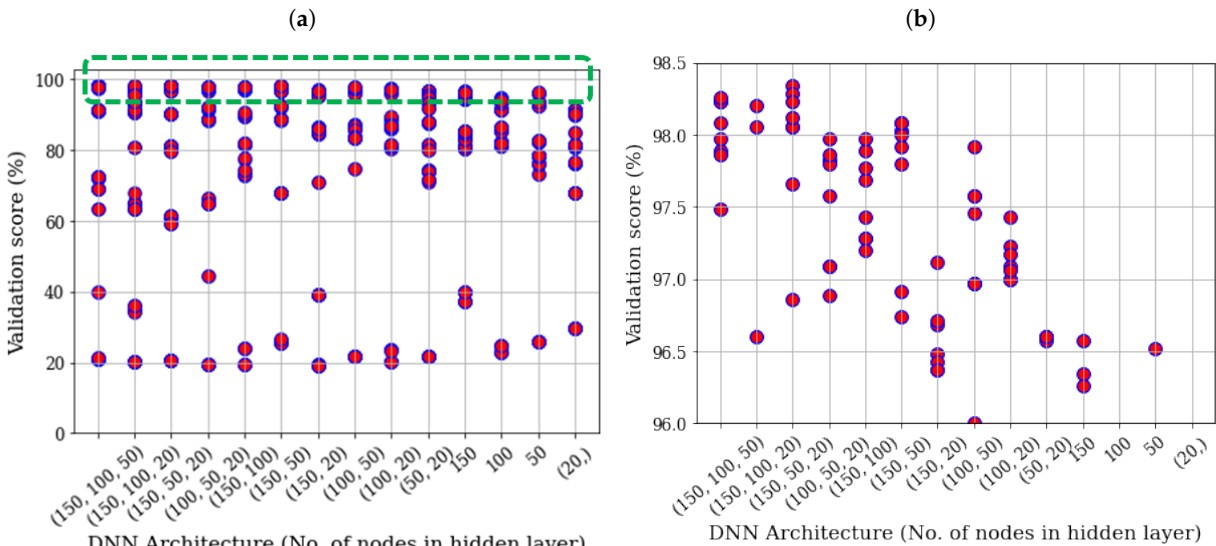

**Figure 6.** Validation scores of different DNN architectures over a five-fold cross validation: (**a**) a complete assessment view and (**b**) a zoomed view of the highest validation scores.

As shown in Figure 6a, different validation scores are returned for different DNN architectures with several overlapping values; however, a closer view of the highest validation scores (highlighted in the green dashed rectangle in Figure 6a) are provided in Figure 6b, where it reveals that the DNN with 3 hidden layers and 150, 50, and 20 neurons in the hidden layers, respectively, $(150, 50, 20)$ is the most accurate with a validation score of 99.34%. Particularly, this DNN's accompanying parameters include a *ReLU* activation function, a regularization parameter of 0.001, an *Adam* weight optimizer, and a constant learning rate of 0.001. This clearly verifies the superior advantage of the *Adam* weight optimizer, lower learning rate (with small regularization), and the *ReLU* activation function for reliable diagnostic performance. Although an MLP would train faster, the results in Figure 6 reveals the comparative poor learning efficiencies of such fast learning architectures, which may preform even poorer with bigger learning rates and weight optimizer.

As the study proposes, the cost-aware diagnostic framework demands the need for cost-efficient modules in place while considering the accuracy of the model. Thus far, the grid search provides a paradigm for optimal DNN architecture and parameter selection; however, the early stopping strategy provides an extra avenue for minimizing the training time by stopping the training/validation process as soon as a decline or steady validation score is experienced. Figure 7 shows the training process with the early stopping strategy.

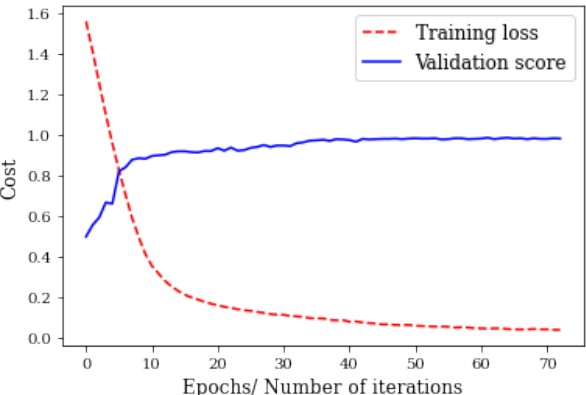

**Figure 7.** DNN training/validation process over 160 iterations.

As observed, although a command of 500 epoch was initiated for the model, it is optimally validated at about the 73rd epoch. This hints that the cost efficiency of the model while also being accurate. Next, the test data was employed on the trained model in an unsupervised manner for testing. Consequently, the test performance of 99.23% accuracy was achieved after a five-fold cross-validation.

To visualize the classification performance, a standard locally linear embedding (LLE) algorithm [4] ($NN = 100$) was used to reduce the features to a two-dimensional feature vector for visualization, Figure 8 shows the isolation of the fault conditions where the axes represent the first and second embedding ($LLE-1$ and $LLE-2$) of the 13 input features.

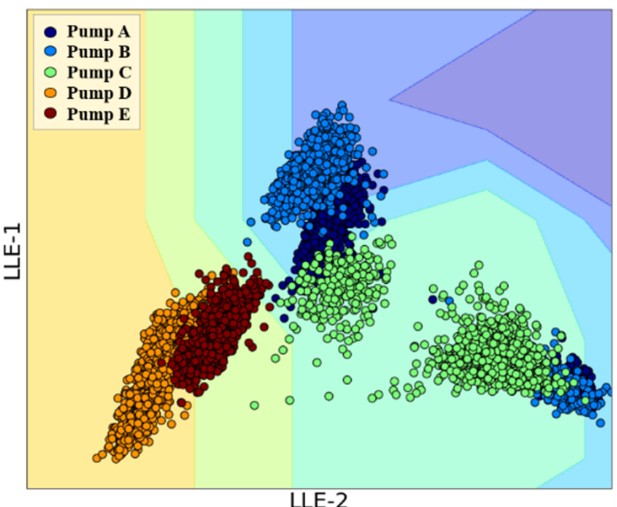

**Figure 8.** Fault isolation by DNN using LLE ($NN = 100$).

As shown, there is quite a reasonable level of discriminance in the features for each of the operating conditions. As observed, there would be a higher probability of false positives for pumps B and C due to the overlapping effects shown in Figure 8 (*far right*). This provides a visual intuition of anticipated isolation results and may also motivate continued studies for more discriminative feature extraction.

### 4.3. FDI Performance Evaluation

Performance metrics for monitoring, fault detection, and diagnostic systems are well established, including accuracy, precision, sensitivity, recall, F1-score, support, etc. These metrics, at zero false alarm rate, return a value of 1 while, at a 100% false prediction, returns a value of zero. Using the standard classification evaluation metrics presented in Equations (5)–(9), their results are summarized in Table 5.

**Table 5.** FDI performance evaluation of DNN.

| Pump | Accuracy | Precision | Recall | F1-Score |
|------|----------|-----------|--------|----------|
| A | 97.9% | 96.1% | 96.1% | 96.1% |
| B | 98.1% | 96.3% | 96.2% | 96.1% |
| C | 99.7% | 99.0% | 99.0% | 99.1% |
| D | 100% | 100% | 100% | 100% |
| E | 100% | 100% | 100% | 100% |

To further assess the performance of the diagnostics framework, the confusion matrix is presented in Figure 9.

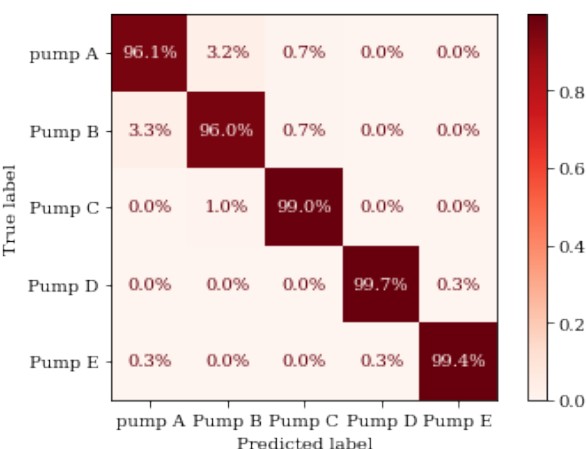

**Figure 9.** FDI evaluation: Confusion matrix.

As shown, the model returns the lowest FPs and FNs for pumps C, D, and E with TP values of 99.0%, 99.7%, and 99.4%, respectively, whereas Pumps A and B returned the highest FPs, as observed down the first, second, and third columns with 3.6% (3.3% + 0.3%), 4.2% (3.2% + 1.0%), and 1.4% (0.7% + 0.7%). The FNs are observed across the first, second, and third rows with 3.9% (3.2% + 0.7%), 4.0% (3.3% + 0.7%), 1.0%, 0.3%, and 0.6% (0.3% + 0.3%) for pumps A–E, respectively. As much as the model was able to diagnose all the classes (Pumps) accurately, as observed, the false alarm rate is quite minimal and with relatively few FNs and FPs for each class, the proposed framework is validated.

In addition, the cost-efficiency of the proposed methodology was assessed by comparing the computational times and accuracy in the prediction of the algorithm in several scenarios. First, the algorithm was employed on all the extracted features and again employed on only spectral domain and time domain features over 500 iterations with an early stopping criteria. Next, the respectively trained models are employed for testing on the test data set. Figure 10 shows the computational cost assessments in these scenarios.

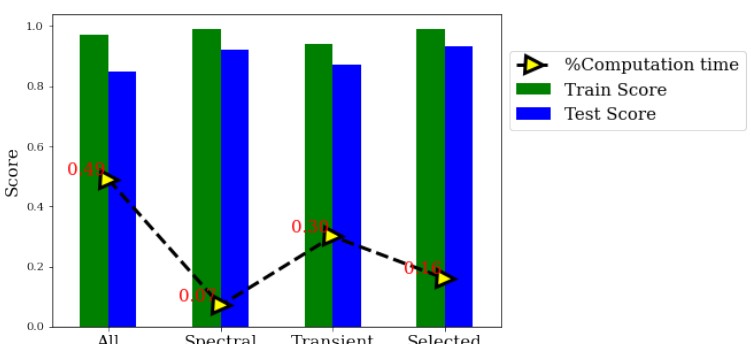

**Figure 10.** Cost assessment for the proposed model.

The green and blue bars represent the train and test scores of the DNN model in each of the cases, respectively, while the yellow triangles represent the percentage (in seconds) of the overall computation time (training and testing) for each of the scenarios. As shown, using all the features is the most cost-inefficient approach as it returns the least training and test scores (91.3% and 84.8%, respectively) while constituting the highest computational time (about 49% of the whole computational time). On the other hand, the use of the spectral features seems to be the most cost-efficient with a cost of 7%; however, this can be attributed to its fewer dimension (only four spectral features). With test scores of 92% and 93.4% for using spectral features and using discriminant features (selected features), respectively, the accuracy of the proposed approach surpasses the use of only spectral

features at a computational cost of about 16% of the whole computation process. On the downside too, using only transient features is just as less optimal as using the whole feature set with a 30% cost in overall computation time. The proposed FDI technology not only ensures that the salient fault features are selected but also further minimizes the cost of computation for real-life applications.

### 4.4. Comparative Assessments with Other Models

To further assess the model's diagnostic performances, other popular ML-based models such as the support vector machine (SVM), Gradient Boosting Classifier (GBC), Gaussian Process Classifier (GPC), Naive Bayes classifier (NBC), etc. were employed alongside the DNN model on the selected features for training and testing, respectively. Table 6 summarizes the test accuracy, precision, recall, F1-score, and computational cost (in seconds) for the respective models.

**Table 6.** Diagnostic performance comparison of ML models.

| Algorithm | Accuracy(%) | Precision(%) | Recall(%) | F1-Score(%) | Cost (Secs) |
|---|---|---|---|---|---|
| Logistic Regression | 96.13 | 96.04 | 96.05 | 96.03 | 0.163091 |
| Nearest Neighbor | 98.53 | 98.51 | 98.51 | 98.50 | 0.010305 |
| Linear SVM | 97.73 | 97.68 | 97.68 | 97.68 | 0.074509 |
| Gaussian SVM | 98.67 | 98.63 | 98.63 | 98.63 | 0.092407 |
| GBC | 94.83 | 98.98 | 98.97 | 98.97 | 57.663607 |
| Decision Tree | 98.07 | 98.02 | 98.01 | 98.02 | 0.022450 |
| Random Forest | 98.80 | 98.77 | 98.77 | 98.77 | 4.692902 |
| DNN | 99.24 | 98.73 | 98.94 | 99.23 | 4.803478 |
| NBC | 93.33 | 93.52 | 93.16 | 93.25 | 0.002171 |
| AdaBoost classifier | 58.60 | 67.51 | 59.39 | 56.82 | 0.347029 |
| GPC | 96.87 | 96.81 | 96.81 | 96.77 | 42.794803 |

As shown, the DNN model outperforms the other models based on test accuracy; however, it would be illogical to assume that this performance superiority is generalized since model parametrization was not implemented for the other models (default parameters were used for the other models). Nonetheless, from the a high computationally time of 57.66 and 42.8 s from the GBC and GPC, respectively, even with high test accuracies, it is clear that the GBC and GPC are are quite computationally expensive and should not suffice in a cost-aware framework. In sharp contrast, algorithms such as the NBC, AdaBoost Classifier, Decision Tree, SVM, Nearest neighbor, and Logistic regression are quite computationally inexpensive, as observed from their respectively small computation costs.

Interestingly, the Random forest classifier and the proposed DNN model both stay within fairly acceptable computational costs (less than 5 s); however, from the relatively lower accuracy (a difference of 0.44%), the proposed model ranks higher in the scale of preference. This comparison hints at providing a paradigm for choice of classifier depending on the metric of interest. These are more preferred in most cost-aware applications where accuracy is also desired.

### 5. Conclusions and Continued Works

Accurate vibration monitoring and fault detection/isolation require reliable feature extraction and selection of salient/discriminative features. Statistical features from vibrational signals provide reliable discriminative characteristics in the pumps. By selecting highly discriminant features using reliable methods, accurate fault isolation can be achieved while minimizing the confusion tendencies of the classifier.

This study presents a reliable vibration-based diagnostics approach for solenoid pumps based on statistical features and a DNN {5–(150, 50, 20)–5} classifier following a grid search of optimal parameters and architectures. The results show that the proposed

diagnostics scheme is accurate with a remarkable overall test accuracy of 99.24% and with minimal false alarm rate; however, among the five failure modes presented in this study, Pump B (high viscosity failure mode) showed the lowest fault isolation performance with an accuracy of 99.24%, a precision of 98.73%, a recall of 98.94%, and an F-1 score of 99.23%.

Future research shall be aimed at obtaining more experimental data to cover other failure modes for a more comprehensive diagnostic scheme. In addition, ongoing studies are focused on developing a prognostics scheme for the remaining useful life precision of the pumps using deep learning methods after a run-to-failure test. In addition, since the feature selection threshold was chosen based on human experience, the efficiency of the proposed diagnostic scheme relies significantly on the operator/user. This presents a reasonable amount of room for human errors/limitations in the diagnostic performance; consequently, our ongoing research is aimed at employing meta-heuristic search algorithms such as genetic algorithm, particle swarm optimization (PSO), ant colony optimization, etc. for automatically choosing the appropriate threshold value in an unsupervised manner. Our concerns in this automated selection approach is the increased computational costs associated with the meta-heuristic algorithm when integrated with the diagnostic framework. Sadly, the cost awareness that is a significant evaluation criteria for the acceptability of the proposed study would be jeopardized.

**Author Contributions:** Conceptualization, S.K. and U.E.A.; methodology, S.K. and U.E.A.; software, U.E.A.; formal analysis, U.E.A.; investigation, S.K. and U.E.A; resources, S.K., U.E.A., and J.W.H.; data curation, S.K. and U.E.A.; writing—-original draft, S.K.; writing—review and editing, U.E.A.; visualization, U.E.A.; supervision, J.-W.H.; project administration, J.-W.H.; funding acquisition, J.-W.H. All authors have read and agreed to the published version of the manuscript.

**Funding:** This work was supported by the National Research Foundation of Korea (NRF) grant funded by the Korea government (MIST) (No. 2019R1/1A3A01063935).

**Data Availability Statement:** The data presented in this study are available on request from the corresponding author. The data are not publicly available due to laboratory regulations.

**Conflicts of Interest:** The authors declare no conflict of interest.

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
