# Peer review of "A Cost-Aware DNN-Based FDI Technology for Solenoid Pumps"

_electronics, doi:10.3390/electronics10192323_

Round 1
Reviewer 1 Report
Summary:
This paper proposes a cost-efficient FDI technology for solenoid pumps based on spectral and transient diagnostic information in the application of Fluid Pumps. I appreciate the contributions and believe the paper is well-prepared but still has some concerns to be addressed. Detailed comments are listed below.
Pros:
- The solution is reasonable and seems effective according to the experiments.
- The paper is generally well-written and easy to follow.
Cons:
Major
-In Abstract, the proposed model should be more detailed.
- I wonder about the design of DNN architecture. Why do you select a 3-layer DNN classifier, an architecture of {5–[150, 100, 50]–5}? Please show parameter sensitivity analysis on number of layers and number of neurons.
- The related work about deep learning, solenoid pump FDI and prognostics is not enough.
- Why do you leverage MLP, rather than other machine learning methods, like SVM, Decision Tree, Random Forest? What are their results?
Minor
- Description of MLP seems redundant.
- I recommend authors introduce the background much clearer.
- Some deep-learning-based work should be added. Multi-camera multi-player tracking with deep player identification in sports video (Pattern Recognition), Deep-IRTarget: An Automatic Target Detector in Infrared Imagery using Dual-domain Feature Extraction and Allocation(IEEE Transactions on Multimedia), Deep-learning-based burned area mapping using the synergy of Sentinel-1&2 data(Remote Sensing of Environment).
Reviewer 2 Report
The paper addresses an interesting and actual problem that refers to the fault diagnosis based on signal analysis and the use of deep neural networks. The paper is well organised and easy to read. However, there are some aspects to be further clarified. In the following, there are some comments and questions that might help you improve the paper in clarity.
It is not clear if you have acquired the signals in one day or in several days to consider the environment disturbances that might influence the pumps functioning. Also, it is not clear if you have extracted the features for the whole acquired signals or for several windows. If you have divided the acquired signal in windows, then some other information should be given, such as: how many windows have you used, how many samples per window have you considered, what windowing filter have you used.
Regarding the DNN’s design, it is not clear how many inputs the DNN classifier has, how many training samples have you used, how many samples have you considered for mini batches? For how many epochs have you trained the network? Maybe a comparative study with a classic MLP would help understanding the benefits you get by using a net with lots of neurons in its hidden layers.
Please explain all the used abbreviations (DNN is not explained).
Round 2
Reviewer 1 Report
I agree the manuscript could be accepted.